# Rapid ecosystem-scale consequences of acute deoxygenation on a Caribbean coral reef

Maggie D. Johnson [1,2,3,7] ✉, Jarrod J. Scott[1,7], Matthieu Leray [1,7], Noelle Lucey [1,7], Lucia M. Rodriguez Bravo[1,4], William L. Wied[1,5] & Andrew H. Altieri[1,6]

Loss of oxygen in the global ocean is accelerating due to climate change and eutrophication, but how acute deoxygenation events affect tropical marine ecosystems remains poorly understood. Here we integrate analyses of coral reef benthic communities with microbial community sequencing to show how a deoxygenation event rapidly altered benthic community composition and microbial assemblages in a shallow tropical reef ecosystem. Conditions associated with the event precipitated coral bleaching and mass mortality, causing a 50% loss of live coral and a shift in the benthic community that persisted a year later. Conversely, the unique taxonomic and functional profile of hypoxia-associated microbes rapidly reverted to a normoxic assemblage one month after the event. The decoupling of ecological trajectories among these major functional groups following an acute event emphasizes the need to incorporate deoxygenation as an emerging stressor into coral reef research and management plans to combat escalating threats to reef persistence.

[1] Smithsonian Tropical Research Institute, Balboa, Ancon, Republic of Panama. [2] Tennenbaum Marine Observatories Network, MarineGEO, Smithsonian Institution, Edgewater, MD, USA. [3] Present address: Biology Department, Woods Hole Oceanographic Institution, Woods Hole, MA, USA. [4] Present address: Facultad de Ciencias Marinas, Universidad Autónoma de Baja California, Ensenada, Mexico. [5] Present address: Department of Biological Sciences, Center for Coastal Oceans Research, Florida International University, Miami, FL, USA. [6] Present address: Department of Environmental Engineering Sciences, University of Florida, Gainesville, FL, USA. [7] These authors contributed equally: Maggie D. Johnson, Jarrod J. Scott, Matthieu Leray, Noelle Lucey.
✉email: magjohnson@gmail.com

Acute deoxygenation events, and the ensuing mass mortality and formation of dead zones in coastal ecosystems, are increasing in frequency and severity due to anthropogenic impacts including eutrophication and climate change[1,2]. Hypoxic events have been well studied in temperate systems, where sudden deoxygenation has caused ecosystem collapse and loss of goods and services[3]. However, the effects of analogous events in the tropics remain poorly understood[2], despite the occurrence of physical characteristics of coral reef geomorphology (e.g., semi-enclosed bays) and hydrology (e.g., restricted water flow) that exacerbate deoxygenation, particularly when coupled with local and global human impacts[4]. Though more than 10% of coral reefs globally are at elevated risk of hypoxia[4], global change research in these habitats has largely focused on the singular effects of warming and ocean acidification[5,6].

Deoxygenation has long been recognized as a leading threat to coastal marine ecosystems[3], but has only recently emerged as a critical threat in the tropics[2]. Tropical coral reefs are ecologically and economically valuable ecosystems in global decline due to decades of accumulating environmental impacts[7,8]. The physical framework of coral reefs is created by growth and biogenic calcification of scleractinian corals. As a result, the reef habitat and the ecosystem services they support are directly threatened by the loss of living corals[9]. The survival of reef-building corals is tied to the performance of microorganisms, including their endosymbiotic dinoflagellates[10] and the reef-associated microbial assemblages that are responsible for cycling nutrients and organic materials that facilitate energy transfer and trophic linkages through the microbial loop[11]. However, the integrity of these ecological networks and their emergent ecological functions appear vulnerable to environmental change[12,13].

Relatively little is known about the ecosystem-scale effects of acute deoxygenation (i.e., hypoxic) events and subsequent recovery dynamics on coral reefs, in part because episodes of hypoxia typically last just a few days, and oftentimes go undocumented. Here, deoxygenation and hypoxia refer to depleted dissolved oxygen (DO) concentrations in seawater, from just above anoxia (complete oxygen depletion, $0 \, \text{mg} \, \text{l}^{-1}$) to just below normoxia (100% air saturation, or $6.8 \, \text{mg} \, \text{l}^{-1}$ at 27 °C)[3], though the typical threshold used to denote hypoxic conditions is $\leq 2 \, \text{mg} \, \text{l}^{-1}$[1]. On some coral reefs, respiratory depletion of oxygen at night can cause DO to approach hypoxia over a diel cycle[14]. However, the magnitude of oxygen depletion that occurs during an acute deoxygenation event, as well as the duration of exposure, are vastly different than what organisms experience during a typical diel cycle[15]. Indeed, recent evidence demonstrates that prolonged exposure of acute deoxygenation events can decimate coral communities in a matter of days[4,16].

The first well-documented assessment of hypoxia on a coral reef followed an event in late September, 2010 in Bocas del Toro on the Caribbean coast of Panama[4]. Formation of hypoxic conditions was likely related to a period of low wind activity, warmer water temperatures corresponding to the end of the boreal summer, and high levels of eutrophication and organic pollution in Bahía Almirante[4]. The 2010 event resulted in mass mortality of virtually all corals at a depth from 10–12 m down to the bottom of reefs at ~15 m at affected sites[4]. A handful of other studies have implicated acute hypoxic conditions in coral reef mortality (reviewed in[17]), but there has yet to be a study that quantifies both abiotic and biotic community responses during a hypoxic event. Here we comprehensively evaluate the environmental conditions and biological responses that occurred during an unfolding hypoxic event on a shallow coral reef, along with community trajectories following a recovery period. By exploring impacts on reef-associated microbes, reef-building corals, and benthic community structure, we show that reef functional groups demonstrate different patterns of resilience to an acute deoxygenation event. Although microbial assemblages quickly recovered, the impacts of coral mortality and benthic assemblage shifts resulting from the event persisted more than a year later. These divergent functional group responses have direct implications for the ability of tropical ecosystems to persist and recover under accelerating environmental change.

## Results

**Characteristics of an acute deoxygenation event**. An acute deoxygenation event occurred again in Bahía Almirante on the Caribbean coast of Panama (Fig. 1) in 2017, 7 years after the first reported hypoxic event in the area[4]. A distinct sharp gradient of DO, or oxycline, was detected in late September on a representative inner-bay coral reef (Cayo Roldan, hereafter impacted site) at 3–4 m depth (Fig. 1d), which likely resulted from the shoaling of hypoxic water that can persist in the water column adjacent to reef habitats at depths below 20 m in Bahía Almirante[15]. Below this shallow oxycline, the benthos was littered with dead, but intact sponges and exposed, moribund, and dying mobile invertebrates (Fig. 1e), as well as hypoxia-associated microbial mats (Fig. 1b). Coral bleaching, tissue loss, and mortality were apparent and widespread within the hypoxic water (Fig. 1a, c). Although the exact duration of the event is unknown, hypoxic conditions were detected on the shallow reef over a period spanning at least 6 days (September 20–26, 2017).

To evaluate the extent and severity of hypoxia and the associated environmental conditions, we conducted a region-wide survey 6 days after the onset of the event. Depth profiles were conducted during the day at 83 sites across the bay with a multiparameter sonde (Yellow Springs Instruments EXO2 measuring DO, temperature, chlorophyll, salinity, pH). Within Bahía Almirante, hypoxic conditions were consistently most severe, and the water column most stratified, farthest from the inlets where bay water exchanges with oxygenated water from the open ocean (Fig. 1d). We selected two focal sites from across this gradient for targeted coral collections and surveys. At the impacted site, DO concentrations in the first meter above the benthos were suboxic ($0.18 \pm 0.12 \, \text{SD} \, \text{mg} \, \text{l}^{-1}$). This was an order of magnitude lower than at the reference site outside Bahía Almirante ($5.47 \pm 0.07 \, \text{SD} \, \text{mg} \, \text{l}^{-1}$) where reefs are well flushed with open ocean water (Cayo Coral, hereafter reference site) (Fig. 1d). The difference in environmental conditions was greatest for DO, with a maximum range in concentrations of $6.57 \, \text{mg} \, \text{l}^{-1}$. Other parameters also varied between the focal sites, but the maximum range between sites was minimal for pH (0.72 units), temperature (1.7 °C), salinity (2.27 practical salinity units), and chlorophyll (2.07 relative fluorescent units), relative to the changes in DO (Supplementary Fig. 1).

**Coral responses to an unfolding deoxygenation event**. Understanding the impacts of an acute deoxygenation event on the dominant coral species in a community is essential to characterizing the short and long-term implications on the broader reef ecosystem. To address this, we quantified how conditions associated with the 2017 event led to bleaching in the lettuce coral *Agaricia tenuifolia* (hereafter *Agaricia*) (Fig. 1a). Bleaching is the process by which corals expel their endosymbiotic algae. Though bleaching is typically observed in response to warming[18], it broadly indicates physiological stress and has been detected in response to other environmental changes including acidification[19] and hypoxia in the lab[20]. *Agaricia* is a representative Caribbean coral species, abundant on the shallow reefs of Bocas del Toro at both the impacted and reference sites at 3–5 m depth[21].

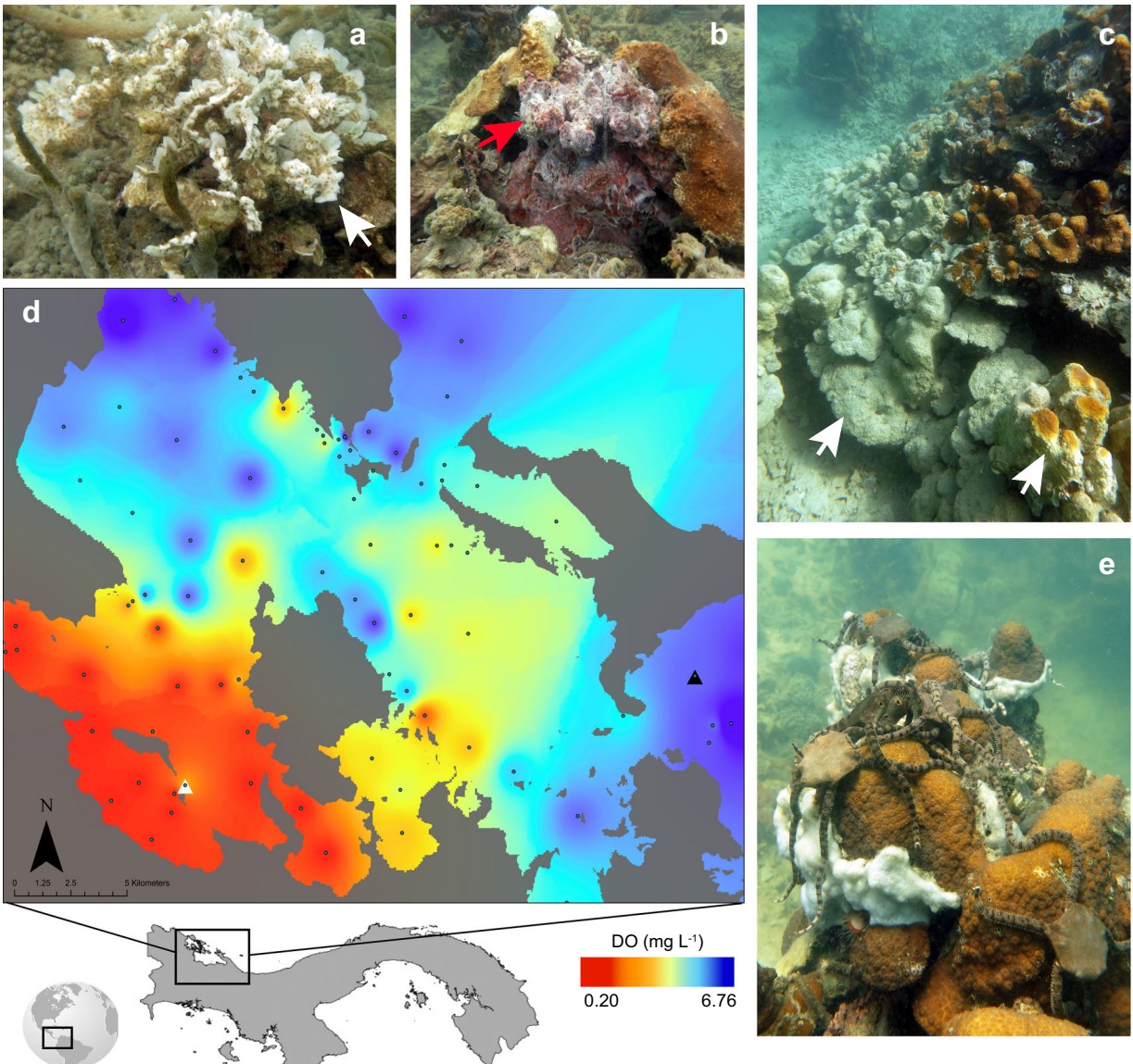

**Fig. 1 Dissolved oxygen concentrations and responses of coral reef taxa during the 2017 deoxygenation event in Bocas del Toro, Panama. a** Coral bleaching (white arrow) was evident in corals at the impacted site (Cayo Roldan, white triangle) during the 2017 event, including the lettuce coral *Agaricia tenuifolia*. Below the oxycline in hypoxic waters, **b** microbial mats diagnostic of anoxic conditions, comprised of thin, white filaments (red arrow), were abundant, and **c** coral bleaching typically progressed from the bottom to the top of boulder corals (white arrow). **d** Map of Bahía Almirante illustrates DO concentrations at the benthos 6 days after initial detection of the event (depths ~3–20 m), with sampling stations noted as black dots. An unaffected coral reef (Cayo Coral, black triangle) located outside Bahía Almirante was used as a reference site. **e** Below the oxycline at the impacted site, invertebrates, including cryptic brittle stars, congregated on the tops of boulder corals.

During the event, we collected three coral fragments from each of six distinct colonies at each site to quantify coral bleaching and photophysiology ($N = 18$ total fragments per site). Colonies (>10 m apart) and fragments were selected haphazardly at each site, although only corals with evidence of remaining live tissue were chosen at the impacted site. Corals from the impacted site were significantly more bleached than their counterparts from the reference site. Symbiont densities, a direct measure of coral bleaching, were 78% lower at the impacted site (linear mixed effects model (LME, $p < 0.001$) (Fig. 2a). Maximum quantum yield ($F_v/F_m$), a measure of photosynthetic efficiency of endosymbiotic algae and a functional indicator of coral bleaching[22], was 12% lower at the impacted site (LME), $p < 0.001$) (Supplementary Fig. 2), and photosynthetic pigments were 59% (chl a) and 56% (chl c) lower (LME, chl a $p < 0.001$, chl c $p < 0.001$) (Supplementary Fig. 2). Together, these physiological metrics revealed

how conditions at the impacted site facilitated coral bleaching, which presaged coral mortality and a persistent shift in benthic community structure. Benthic surveys conducted immediately before the event found no evidence of coral bleaching at the impacted site, which supports the observation that bleaching and mortality were directly associated with event conditions.

**Changes in benthic community structure**. Tracking the macroscopic benthic community during and after a massive perturbation provides insight to the fate of reef communities, and illustrates the scope for potential changes in ecosystem structure and function in a rapidly changing environment. To quantify impacts of the event on the reef benthos, we followed the response of the shallow reef community at the upper edge of the oxycline (3–5 m depth), which was dominated by three branching

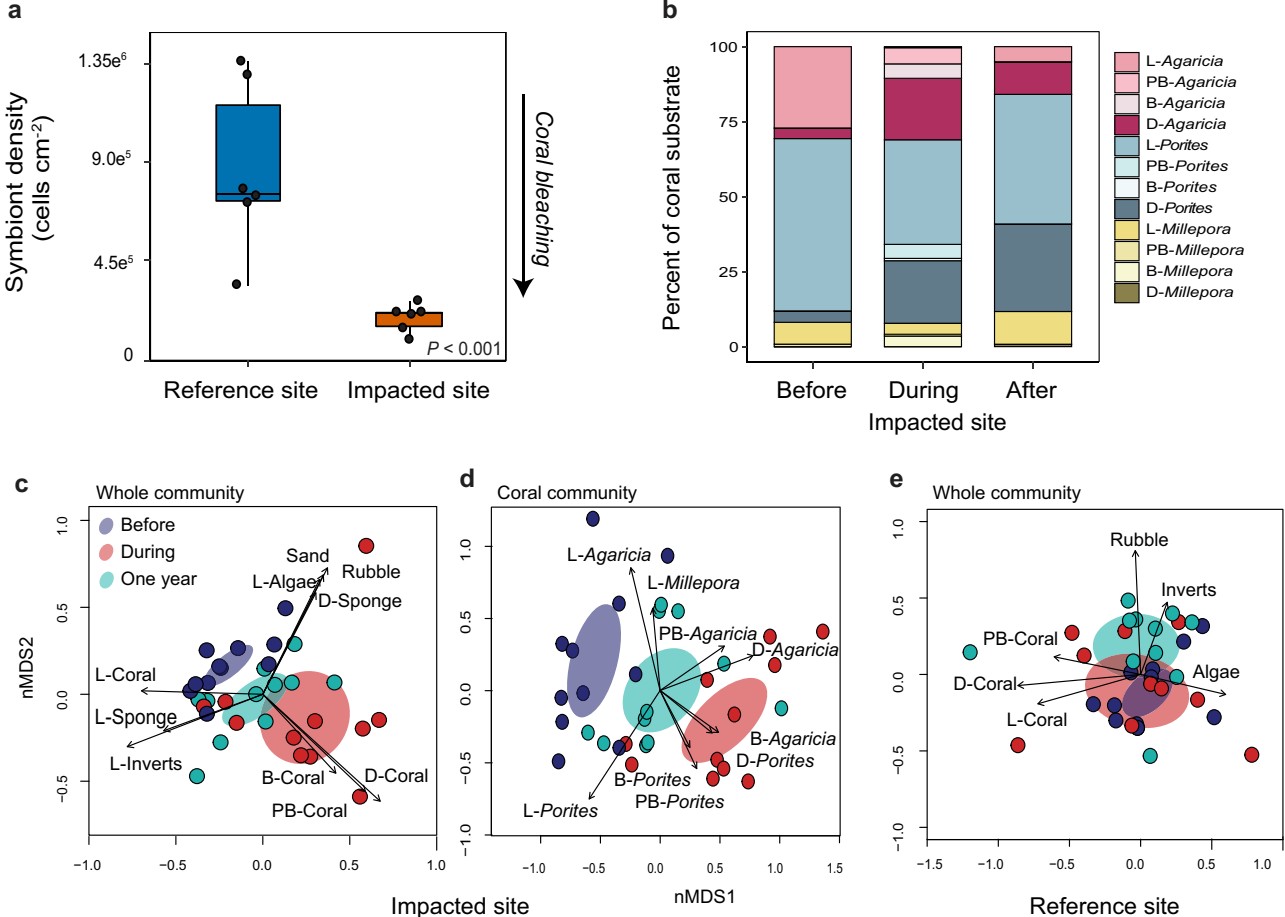

**Fig. 2 Deoxygenation event effects on coral bleaching and community structure. a** Box plot of *Agaricia tenuifolia* symbiont densities during the deoxygenation event, with colony means overlaid as black dots ($N = 6$ biologically independent colony replicates). The center line indicates the median, the box limits the interquartile range, and the whiskers the minimum and maximum values. The black arrow indicates more coral bleaching with decreasing symbiont densities. Differences between sites were evaluated with linear mixed effects models with site as a fixed effect and colony as a random effect ($p = 2.143e^{-08}$). **b** Percent cover of all coral substrate for each coral genus and health condition (L—live, B—bleached, PB—partially bleached, D—dead) at the impacted site 4 months before, during, and 1 year after the event from permanent photoquadrat surveys. **c** nMDS plots showing benthic community assemblages at the impacted site for all functional groups and **d** for only the coral assemblage. Ellipses indicate the 95% confidence interval around the mean assemblage before (blue), during (red), and after (green) the event. Vectors indicate significant contributions of functional groups ($p < 0.05$). **e** nMDS of the benthic community at the reference site, showing no change during the study.

genera: *Porites* sp., *Agaricia tenuifolia*, and *Millepora alcicornis*[23]. Note that species identifications of branching *Porites* are difficult without genetic sequencing[24], thus we refer to individuals in this species complex as *Porites* spp. (hereafter *Porites*). We monitored the benthic community (e.g., macroscopic dominant space holders including corals, sponges, and algae) at the impacted and reference sites 4 months before, during, and 1 year after the hypoxic event using permanent photoquadrats ($N = 11$ per site), and then evaluated changes in community structure at each site over those intervals with a permutational analysis of variance (PERMANOVA) and pairwise contrasts.

A dramatic and long-lasting shift in benthic community structure was initiated during the hypoxic event (PERMANOVA, $p < 0.001$) (Fig. 2b–d). Live coral cover decreased by 50%, from an average (±SD) of $30 \pm 9.4\%$ before to $15 \pm 8.9\%$ during the event, which corresponded with an increase in dead and bleached corals. One year later, the benthic assemblage had not recovered (Adonis, $p = 0.033$), with live coral cover remaining lower than before the event at $23 \pm 10.7\%$. Conversely, the benthic assemblage at the reference site did not change over time (PERMANOVA, $p = 0.901$) (Fig. 2e).

To explore contributions of coral taxa in different conditions of health (live—L, dead—D, bleached—B, partially bleached—PB) to the observed shifts in overall benthic assemblage, we conducted community analyses within only the coral assemblage at the impacted site. The coral community appeared to drive the shifts in overall benthic community structure at the impacted site over time (PERMANOVA, $p < 0.001$) and failed to recover a year later (Adonis, $p = 0.012$) (Fig. 2d). These changes were in turn driven by shifts in the relative abundance of live, bleached, and dead corals of the two dominant genera, which together accounted for >90% of the coral habitat (Supplementary Fig. 3). Initially, live *Agaricia* comprised 27% of total available coral habitat (dead = 3%, bleached = 0%) and live *Porites* comprised 58% (dead = 3%, bleached = 0%) (Fig. 2b). During the event, live *Agaricia* decreased to <1% of total coral habitat and *Porites* to 32%, while dead *Agaricia* and *Porites* accounted for 22% and 19%, and bleached accounted for 12% and 7% for each taxon, respectively (full and partial bleaching combined). Notably, 1 year after the event there was no evidence of bleaching, and only 5% of available coral habitat was live *Agaricia* and 44% was *Porites*, while 9% was dead *Agaricia* and 28% was dead *Porites* (Fig. 2b).

Our results illustrate how the 2017 event had ecosystem-scale effects in the shallow reef assemblage through coral mortality and shifts in the community structure of benthic macrofauna. The sluggish recovery of live coral cover reveals how a hypoxic event that lasted several days can have lasting consequences for coral reef community structure that directly influence ecosystem functioning.

**Shifts in microbial communities following deoxygenation.** Microorganisms both contribute to and support high biodiversity and productivity on coral reefs[11]. Because microbial communities respond rapidly to physicochemical changes[25], identifying how microbial assemblages shift in response to stress can provide insight to both the consequences of perturbation for micro-organisms and the environmental causes of change[26]. As reef microhabitats host different microbial consortia[27], we focused on the water-borne microbial assemblage at the benthic-pelagic interface to assess an integrated ecosystem response, as opposed to coral or substrate-specific. Microbes were collected in three 1 l water samples from 0.5 m above the benthos at each site during the hypoxic event, and again a month after the hypoxic event. 16S rRNA sequencing revealed a shift in microbial community composition from normoxic to hypoxic conditions, with 111 differentially abundant amplicon sequence variants (ASVs) between hypoxic and normoxic samples (Fig. 3a and Supplementary Data 1). At the broader community level, hypoxic samples showed a marked decrease in Cyanobacteriia, SAR406, and Acidimicrobiia. Notably, a month after the event the microbial communities at the impacted site closely resembled those from the normoxic reference site (Fig. 3a), indicating that the community rapidly rebounded to a normoxic assemblage with the return of oxygenated conditions.

To further explore the microbial assemblages, we conducted shotgun metagenome sequencing on one sample from each site and sampling period ($N = 4$ total). Taxonomic analysis of short reads showed patterns of diversity similar to the 16S rRNA data, specifically an increase of Campylobacterota and Alphaproteobacteria in the hypoxic sample and a decrease of Cyanobacteria (Fig. 3b). Metagenomic analysis also showed a decrease of viral reads in the hypoxic sample (Fig. 3b). Co-assembly of metagenomics reads revealed 44 microbial bins and 92 viral bins ($\geq$ 100 kbp each). We reconstructed five metagenome assembled genomes (MAGs) (Fig. 3b). MAG01 and MAG03, classified in the family Cyanobiaceae (Cyanobacteria), and MAG05 (*Pelagibacter*, SAR11 clade) were common to both hypoxic and normoxic samples. MAG02, an *Arcobacter* (Arcobacteraceae, Campylobacterota), and MAG04, an *Alliiroseovarius* (Rhodobacteraceae, Alphaproteobacteria), were found exclusively in the hypoxic sample (Fig. 3b). A sixth abundant microbial bin (Bin 13, unclassified Nitrincolaceae) was found exclusively in the hypoxic sample. Despite manual bin curation, Bin 13 maintained high redundancy (68%) and could not be confidently resolved to a high-quality MAG (Fig. 3b). It is worth noting that the taxonomy of *Arcobacter* is currently in flux[28] and Pérez-Cataluña et al. recently proposed splitting *Arcobacter* into six genera[29]. Though taxonomic analysis against the Genome Taxonomy Database (GTDB)[30] (see "Methods") classified MAG02 as a *Poseidonibacter*, for clarity, we refer to MAG02 as *Arcobacter* throughout.

We compared MAG02 to 72 publicly available *Arcobacter* and *Sulfurimonas* (outgroup) genomes, and MAG04 to 14 publicly available *Aliiroseovarius* and *Roseovarius* (outgroup) genomes using a phylogenomic approach (hereafter, *Arcobacter* and *Aliiroseovarius* phylogenomic trees). We constructed the *Arcobacter* phylogenomic tree from 17 manually curated single-copy gene clusters (1309 total) and the *Aliiroseovarius* phylogenomic tree from 28 manually curated single-copy gene clusters (532 total). *Arcobacter* genomes grouped into one terrestrial clade, three marine clades, and one clade that was a mix of both environments (Supplementary Fig. 4). MAG02 clustered within a clade of eight marine-associated genomes, predominantly affiliated with marine invertebrate hosts, specifically scallop and abalone. This suggests that MAG02 may normally be host-associated and/or part of the benthic community. The V4–V5 region of the 16s rRNA sequence pulled from the MAG02 assembly 100% matched ASV10 (an unclassified Arcobacteraceae), suggesting that ASV10 and MAG02 are the same organism. MAG04 clustered within a clade of four marine-associated genomes from seawater (*A. marinus*[31] and *A. pelagivivens*[32]), estuarine sediment (*Aliiroseovarius* sp. PrR006[33]), and a sea squirt (*A. halocynthiae*[32]) on the *Aliiroseovarius* phylogenomic tree.

Functional annotations of genes in MAG02 (*Arcobacter*) and MAG04 (*Aliiroseovarius*) indicated a total of 46 and 21 metabolic pathway modules, respectively (Kyoto Encyclopedia of Genes and Genomes, KEGG) with >75% completeness (Supplementary Data 2). MAG02 contained a complete Sox pathway module (M00595). The seven genes of this complex (*soxXYZABCD*) are involved in the oxidation of sulfur-containing thiosulfate to sulfate, a ubiquitous product of anaerobic processes. The Sox pathway is common in sulfur-oxidizing bacteria[34] and *Arcobacter* in particular[35] (we found complete Sox pathways in 63 of the 72 *Arcobacter* genomes in our analysis). Though all *Roseovarius* and *Aliiroseovarius* (save the two *A. crassostreae*) genomes contained a complete Sox pathway, we only detected *SoxD* in MAG04.

MAG02 contained over 70% of the genes from the reverse Krebs cycle (M00173), including a complete pyruvate oxidation (pyruvate to acetyl-CoA) pathway (M00307)—specifically pyruvate ferredoxin oxidoreductase genes (*porABCD*)—pyruvate, water dikinase (K01007), malate dehydrogenase (K00024), fumarate hydratase (K01676), a near-complete fumarate reductase pathway (M00150, *frdABC*), succinate dehydrogenase/fumarate reductase (K00240), 2-oxoglutarate/2-oxoacid ferredoxin oxidoreductase (alpha and beta subunits, plus 2-oxoglutarate ferredoxin oxidoreductase gamma subunit), isocitrate dehydrogenase (K00031), and aconitate hydratase (K01682). MAG02 contained the complete cytochrome c oxidase, *cbb3*-type module (M00156, subunits I–IV). MAG04 contained a near-complete pyruvate oxidation pathway (M00307), specifically pyruvate dehydrogenase genes (*pdhACD*)—but only 46% of the genes from the reverse Krebs cycle. None of the *Roseovarius* or *Aliiroseovarius* genomes contained all genes from the reverse Krebs cycle. However, MAG04 did contain genes from modules for carbon fixation via photosynthesis, including both genes from the Crassulacean acid metabolism (CAM), light module (M00169)—malate dehydrogenase and pyruvate, genes from the reverse Krebs cycle. None of the *Roseovarius* or *Aliiroseovarius* genomes contained all genes from the reverse Krebs cycle. However, MAG04 did contain genes from modules for carbon fixation via photosynthesis, including both genes from the CAM, light module (M00169)—malate dehydrogenase and pyruvate, orthophosphate dikinase. While MAG04 contained only ~65% of the genes from the reductive pentose phosphate cycle (Calvin cycle, M00165)—specifically phosphoglycerate kinase, glyceraldehyde 3-phosphate dehydrogenase (gapA), fructose-bisphosphate aldolase (class I), transketolase, ribose 5-phosphate isomerase A—all other *Roseovarius* and *Aliiroseovarius* contained more complete pathways. MAG04 also contained the complete cytochrome c oxidase, *cbb3*-type module (M00156, subunits I–IV).

These results suggest that MAG02 and MAG04 are capable of growing at low-oxygen concentrations and fixing $CO_2$ using thiosulfate as an electron acceptor. Notably, the *cbb3*-type cytochrome c oxidases, the Sox, pyruvate oxidation, and fumarate

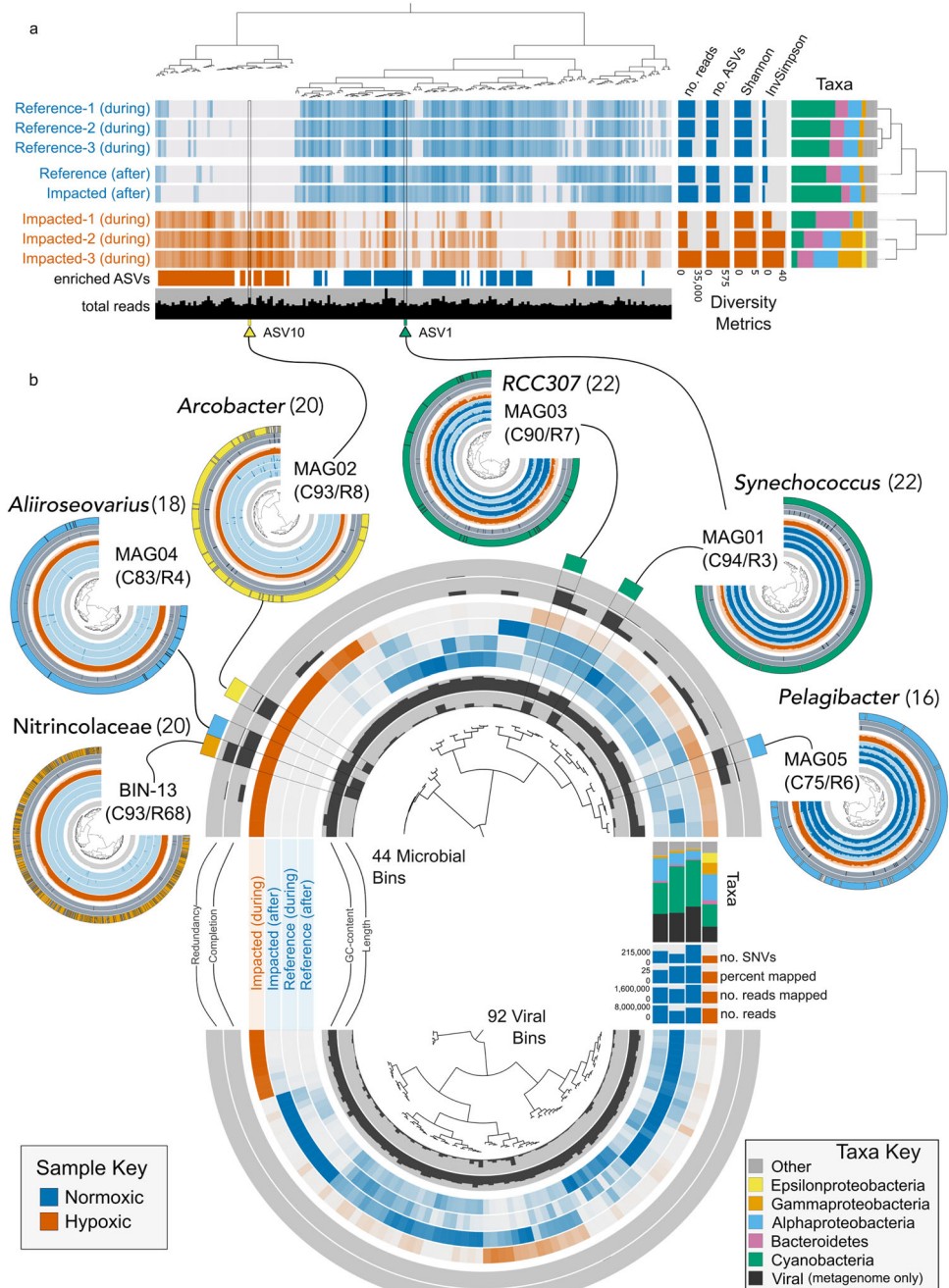

**Fig. 3 Hypoxic and normoxic reef-associated microbial assemblages. a** Community composition from 16S rRNA sequencing of water samples from the impacted (red) and reference sites (blue) during and one month after the deoxygenation event ($N = 3$ per site, after samples were pooled before sequencing). Hierarchical clustering of samples (right dendrogram) and ASVs (top dendrogram) based on Euclidean distance and Ward linkage. Each vertical line represents a unique ASV, color intensity indicates the log-normalized abundance, and no color indicates an ASV was not detected. The colored bar indicates which ASVs were enriched in the hypoxic vs. normoxic conditions as determined by Indicator Species Analysis. Taxonomic profiles show the proportion of major bacterial classes for each sample. **b** Metagenome binning for samples from each site. Each layer represents a sample color-coded by oxygen conditions, and each spoke a distinct bin. Inner and outer layers provide information on the length and GC content, as well as completion and redundancy estimates, for each bin. Bar charts (right) show the total numbers and percent of reads mapped to the co-assembly, the number of single nucleotide variants, and taxonomic content for each sample. The top and bottom semi-circles show the 44 microbial bins and 92 viral bins, respectively. The five MAGs recovered from the assembly are overlaid, including genus-level taxonomic assignments. Bin 13 (Nitrincolaceae) could not be resolved into a MAG but is also included because it was abundant and only detected in the hypoxic samples. Numbers in parentheses indicate the number of ribosomal genes found in each MAG. ASV10 fully matches the 16S rRNA gene sequence retrieved from MAG02 (yellow arrow). Legend keys apply to both figure panels.

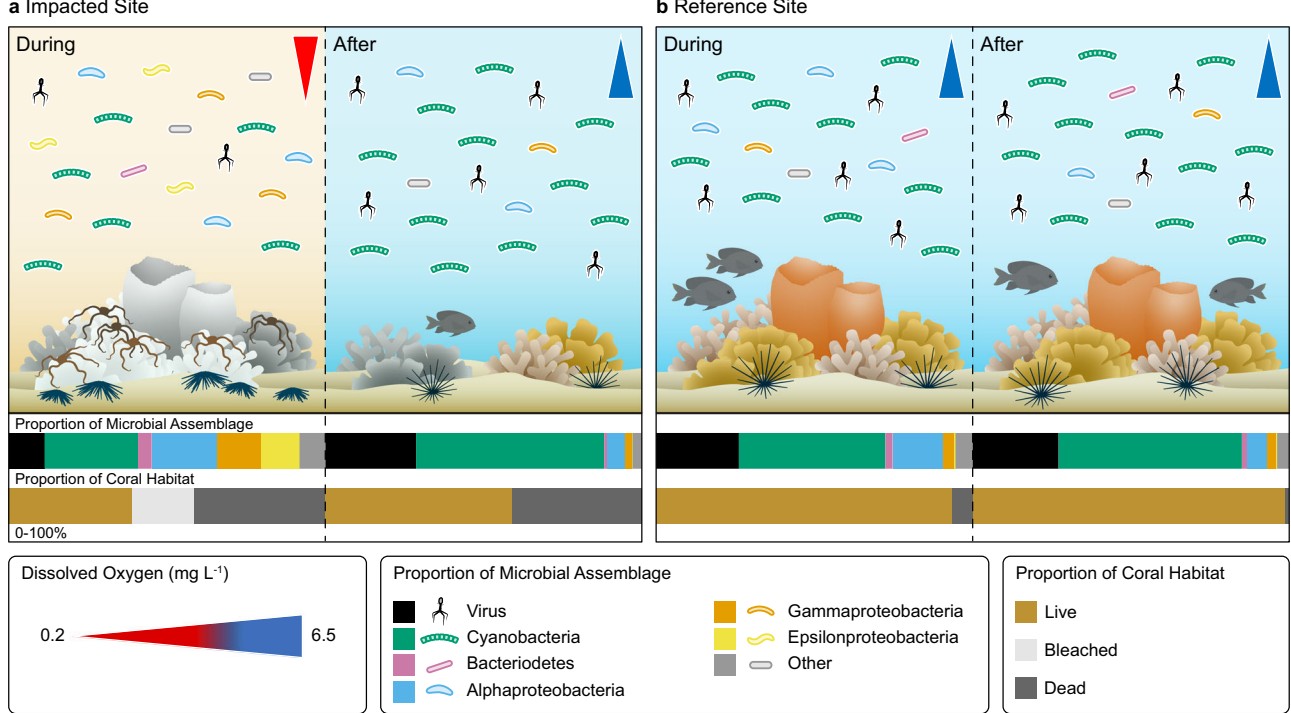

**Fig. 4 Impacts of an acute deoxygenation event on coral reef benthic and microbial community structure.** Conceptual figure depicting how the deoxygenation event shifted microbial and benthic community composition, and how the shift in the benthic assemblage persisted 1 year after the event. In contrast, no such changes were observed at the reference site. Abundance of microbial taxa depicted in the water is proportional to the profiles determined through shotgun metagenome sequencing (top, horizontal stacked bars). The amount of coral habitat is presented as the proportion of coral habitat that was either live (brown), bleached (gray), or dead (dark gray) at each time point based on field surveys (bottom, horizontal stacked bars). Dissolved oxygen concentrations ranged from ~0.2 mg l$^{-1}$ (red) to ~6.5 mg l$^{-1}$ (blue) in hypoxic and normoxic conditions, respectively. Figure created for this paper by Amanda Dillon, Aline Design LLC.

reductase, pathways were completely absent from normoxic communities and detected only in the hypoxic community.

## Discussion

Here we captured the micro-to-macroscopic responses of a shallow community as an acute deoxygenation event unfolded on a Caribbean coral reef (Fig. 4). Conditions associated with the hypoxic event coincided with bleaching, tissue loss, and whole colony mortality in corals at the impacted site. These findings were evident in both physiological measurements collected during the event and visual surveys of coral bleaching and mortality before, during, and after the event. When bleaching is detected on a coral reef, thermal stress is the commonly assumed culprit, and much of our understanding of bleaching in corals has focused on temperature. Our study suggests that the role of oxygen-stress in the field also should be considered in future studies, and supports emerging lab-based evidence that hypoxia can initiate a coral bleaching response[20]. An increase in dead and bleached corals during the event corresponded with a shift in the microbial assemblage to distinct taxa that thrived in hypoxic conditions (e.g., *Arcobacter*). But unlike the macrobenthos, the microbial assemblage rapidly reverted to a normoxic assemblage within a month of the event. The number of studies to implicate hypoxia in mass mortalities on coral reefs is increasing[16,36]. Our study adds to that growing body of work by characterizing a coral bleaching response to in situ conditions associated with an active deoxygenation event and the concurrent responses of the benthic and microbial assemblages. Moreover, we expand our knowledge of the depth range of reef hypoxic events by demonstrating the consequences of hypoxic waters shoaling to such shallow depths on a coral reef.

The shallow reef community appeared to be more resilient to an acute deoxygenation event than the adjacent deeper reef coral community, and may show a greater potential for recovery. We observed that virtually all corals in the deeper reef community (at depths ranging from 10 to 20 m) suffered rapid mortality in 2017 (M.D.J., personal observation). Yet, we found that nearly half of living corals in the shallow reef community survived. During the 2010 event, nearly all corals at and below 10–12 m died on nearby reefs, but the shallow coral reef adjacent to those communities was not affected. While the deeper reef community failed to recover from the 2010 event, the shallow reef community affected by the 2017 event showed signs of some recovery 1 year later. Variation in the response of different reef assemblages to an acute deoxygenation event could be related to stress tolerances of the dominant species in each assemblage. In Bahía Almirante, the reefs that persist along the 3–5 m versus 10–12 m depth profiles vary fundamentally in species composition and abundance. The less diverse shallow assemblage is dominated by more stress-tolerant corals, including *Porites* and *Millepora*, while the deeper assemblage is dominated by boulder corals (e.g., *Orbicella*, *Siderastrea*, and *Montastrea*) and plating corals (*Agaricia lamarcki*)[23,37]. Within the shallow community studied here, *Agaricia* appeared to be more negatively affected than the other abundant corals. Greater hypoxia-sensitivity has been documented for another species of *Agaricia* (*A. lamarcki*) relative to other corals found on deeper reefs. Such differential tolerances can result in a shift toward assemblages composed of more tolerant species rather than outright loss of the coral community following a hypoxic event[4], which may have enabled higher survival in the shallow reef community during the deoxygenation event and facilitated gradual recovery a year later. Higher survival

of coral taxa in shallower environments following these events is critical for reef recovery dynamics because surviving corals may provide a source of larvae to facilitate recolonization and recovery of the deeper reef community[38].

The continued persistence of macroscopic reef communities and their potential for recovery following disturbance are linked to the performance of host-associated and free-living reef microbial assemblages, which are essential to proper ecosystem functioning on a coral reef[11,26]. Despite their importance, marine microbial taxa remain relatively poorly characterized in these systems, as does their response to environmental perturbations. Our study provides insight to the fate of microbial assemblages on a coral reef during an acute deoxygenation event and illustrates how the consequences witnessed at the macroscopic scale also manifest in the microbiota. Reef-associated microbes respond rapidly to changes in physicochemical conditions[25], providing reliable indications of both physical and biological processes in nature. Further, the shift we detected from the hypoxic microbial community to a normoxic assemblage after the event subsided suggests that the recovery trajectory of reef-associated water-borne microbes is independent and decoupled from the benthic macrofauna. This may facilitate the resumption of key microbial processes (e.g., biogeochemical cycling) that influence the recovery of other aspects of the reef community. Notably, we identified an *Arcobacter* (Arcobacteraceae) (MAG02) and an *Alliiroseovarius* (Rhodobacteraceae) (MAG04) with marine origins that were both dominant and unique to hypoxic conditions, but were absent from an analogous assemblage collected at the same site a month later, following the return of normoxic conditions. Members of *Arcobacter*, a group of metabolically versatile bacteria, are able to take advantage of and thrive in anoxic water and sediments[39], including in other hypoxic coastal ecosystems[40,41]. The consistent prevalence of *Arcobacter*-like microbial taxa in hypoxic conditions ranging from our coral reef samples to the Baltic Sea suggest that it could be used as an indicator to facilitate early detection and characterization of ecological impacts of hypoxic conditions that may last for only days[26,42]. Functional annotations of *Alliiroseovarius* MAG04 predict similar metabolic capabilities, yet this is the first report of a bacteria in this group thriving in hypoxic seawater.

Our field-based study evaluates an in-progress hypoxic event on a shallow coral reef. Numerous environmental stressors can co-occur during an acute deoxygenation event, including lower pH, warmer temperatures, and higher surface chlorophyll concentrations[4,15]. It is beyond the scope of this study to identify the role of specific drivers in the observed ecosystem responses. However, prior work suggests that hypoxia alone is sufficient to drive coral mortality, and may have a dominant effect relative to warm temperatures[4,43], while recent lab-based work illustrates how hypoxia can trigger coral bleaching[20]. While there was some variability in other environmental factors at the focal sites, the difference in oxygen concentrations was extreme relative to differences in pH, temperature, and salinity (Supplementary Fig. 1). Controlled experiments that manipulate multiple environmental stressors alone and in combination are necessary to identify the specific role of hypoxia in coral bleaching and mortality, and should be a focus of future work[36].

At a local scale, hypoxic events may pose a more severe threat to coral reefs than the warming events that cause mass bleaching. Though hypoxia may occur at the scale of a bay or lagoon, acute events impact all resident aerobic marine life, can decimate reef ecosystems in a matter of days, and can leave little room for recovery[4]. The often unpredictable and fleeting nature of hypoxic events makes documenting and responding to them difficult. Conversely, thermal bleaching events occur over regional or ocean basin scales, and typically manifest after weeks-to-months of sustained stress, which can allow for mobilization of research and response teams for monitoring and mitigation. While the

frequency of bleaching events is well-understood, less is known about the occurrence of hypoxic events in the tropics. The event we document here is the second major event in Bocas del Toro[4], and adds to ~50 known mass mortality events associated with acute hypoxia in the tropics globally, though that number is likely underestimated by an order of magnitude[4]. Key steps toward improving our understanding of the environmental characteristics and ecosystem implications of hypoxic events are to identify habitats that are at high risk[4] and to incorporate regular and continuous monitoring of DO concentrations[14,36]. Their localized nature suggests that policy and management practices, including managing watershed inputs like eutrophication and raw waste effluent, may be effective in reducing the likelihood that a hypoxia event will occur[3]. Coral reefs are already suffering severe losses in response to a range of environmental stressors, and our study illustrates the importance of incorporating deoxygenation into experimental frameworks and management plans.

Our assessment of an in-progress hypoxic event on a Caribbean coral reef demonstrates how acute deoxygenation (i.e., hypoxic) events can drastically alter ecological structure outside temperate ecosystems, with impacts across multiple levels of ecological organization that persist in the macroscopic reef community. Our study also illustrates how the fleeting nature of hypoxia requires a continuous monitoring program to link lasting changes in coral reef communities to environmental change that may come in the form of ephemeral events. This is essential to enabling rapid scientific responses, developing conservation action plans, and guiding management decisions as the risk of deoxygenation in the tropics continues to climb.

## Methods

**Site description.** This study was conducted in Bocas del Toro on the Caribbean coast of Panama. Bahía Almirante is a semi-enclosed bay, where circulation with the open ocean becomes more limited with increasing distance from each of two inlets (Fig. 1d). Conditions characteristic of a low-oxygen event were detected in Bahía Almirante on September 20, 2017. To evaluate the effects of hypoxic conditions on the coral reef community, we measured environmental conditions and collected coral samples for physiological analyses and water samples for microbial community analyses at a hypoxic and reference site during and after the deoxygenation event. Hypoxic conditions were first detected at the impacted site, Cayo Roldan (9.22023, −82.3231), located at the interior of Bahía Almirante. Cayo Coral (9.254583, −82.125383) was used as our normoxic reference site, located ~30 km away and outside Bahía Almirante where reefs are continuously flushed with oxygenated, open ocean water. Samples were collected 5 days after detection of the event, and bay-wide environmental measurements were taken 6 days after detection.

**Environmental conditions.** Extensive bay-wide sampling was conducted between 800 and 1730 h on September 26, 2017 to evaluate the spatial extent of the acute deoxygenation event[15]. DO, temperature, chlorophyll, salinity, and pH were quantified with a YSI multiparameter sonde with an optical DO sensor (YSI EXO2 and EXO optical DO Smart Sensor; Yellow Springs, USA) at 83 sites throughout the bay. Reported values are the average of multiple measurements from 1 meter above the benthos at each site. Data points were interpolated with inverse distance weighting techniques using ArcGIS to visualize the variation in parameters on the sampling day.

**Coral bleaching measurements.** The reef-building lettuce coral *Agaricia tenuifolia* was collected from each site on September 25, 2017 between 0600 and 1000 h. Scuba divers collected three fragments (~4 cm$^2$) from each of six colonies at both the impacted and reference sites. Corals were stored in a cooler filled with seawater from the respective site and transported to the wet lab facilities at the Smithsonian Tropical Research Institute's (STRI) Bocas del Toro Research Station for physiological analyses. Coral health was assessed by quantifying the maximum quantum yield of each coral fragment using a blue-light Pulse Amplitude Modulated fluorometer (PAM) (Junior Pam, Walz). Samples were dark adapted in the collection cooler for 2 h, with measurements taken midday. Three measurements were taken from remaining living tissue on each fragment at ~1 cm from the edge. Instrument settings were optimized to yield initial fluorescence readings (F$_0$) between 300 and 500 units, while minimizing measuring light intensity to reduce actinic effects. The same settings were used for corals from both sites: measuring light intensity 6, saturation pulse 8, frequency 2, width 0.8, gain 1[44]. The three measurements from each fragment were averaged for one response value per fragment for subsequent analyses. Directly after PAM measurements, coral tissue was stripped with an airbrush and filtered seawater (0.7 μm) from the underlying skeleton and the

resulting blastate was homogenized for 15 s with a handheld electric homogenizer. Samples were frozen at −20 °C for analyses of zooxanthellae (i.e., symbiont) densities and pigment content.

Coral samples were thawed and then processed for symbiont densities and pigment content[45]. The blastate was vortexed and then centrifuged at $1800 \times g$ for 4 min to separate the host coral animal tissue and symbiont fractions. The resulting pellet, containing the symbiotic zooxanthellae cells, was resuspended in filtered seawater, homogenized by vortexing, and separated into aliquots for cell counts and pigment extractions. Counts were performed with a hemocytometer and compound microscope, with six replicate counts that were averaged for each coral sample. Algal symbiont counts were normalized to sample volume and coral surface area, determined by wax dipping[46], and are expressed as cells per $cm^2$. Chlorophyll was extracted from the symbiont fraction of each coral sample in 100% acetone at 20 °C for 4 h[47]. Pigment samples were then centrifuged at $1800 \times g$ for 2 min and analyzed with a Genesys 30 Spectrophotometer at wavelengths of 630, 663, and 750[47]. Chlorophyll concentrations were calculated from the equations of Jeffrey and Humphries[47], normalized to coral surface area, and expressed as µg l$^{-1}$.

Coral response variables were evaluated for normality and homogeneity of variances by visual assessment of residuals, a Shapiro–Wilk test for normality, and Levene's test for equality of variances. Zooxanthellae densities and pigment concentrations were log-transformed to meet assumptions, and models were conducted with transformed data. Untransformed data are presented in figures to aid interpretation. Response variables were analyzed separately with LME models using maximum likelihood, with site as a fixed effect and colony as a random effect, using *lmer* in the package *lme4*[48] (v1.1-26) in the R environment[49] (v4.0.2). Site significance was evaluated with analysis of variance tables generated from type II sum of squares using Satterthwaite degrees of freedom with the package *lmerTest*[50] (v3.1-3).

**Benthic community structure.** Benthic community composition was evaluated 4 months before the event (May 2017), 2 weeks after the event (October 2017, referred to as during), and 1 year after the event (October 2018). Permanent photoquadrat transects at a depth of 4 m were monitored at both the impacted site and reference site. Transects span 50 m, with photographs taken of permanently marked plots ($0.7 \times 1$ m$^2$) every 5 m ($N = 11$). Image analysis was performed in CoralNet by overlaying 50 random, stratified points and identifying the underlying benthos to the finest taxonomic resolution possible. Species were then categorized by functional group.

Benthic community composition was analyzed separately for each site across the three time points with a PERMANOVA with time point as a fixed factor using adonis2 in the R package *vegan*[51] (v2.5-7), and pairwise contrasts were accomplished with *pairwise.adonis*[52]. Multivariate differences in functional group community composition across time points were visualized with nonmetric multidimensional scaling, with vectors representing significant correlations with axes at $p < 0.05$.

**Microbial community sampling and processing.** Three seawater samples were collected in sterile Whirl-Pak Bags from ~0.5 meter above the reef at both the impacted and reference sites on September 25, 2017 (during hypoxia) and on October 20, 2017 (1 month after) for microbial community analyses. Samples collected during the event were analyzed separately, while post-event samples were pooled prior to sequencing. A separate water sample was collected at each site and time point for metagenomic analysis. Water samples were kept on ice and in the dark until filtration at the laboratory, where they were then vacuum filtered through 0.22 µm nitrocellulose membranes (Millipore). Filters were frozen and transported to STRI's molecular facility at Isla Naos Laboratory in Panama City within 6 h of filtration, and stored at −80 °C until DNA extractions. DNA was extracted from each filter using a Qiagen Powersoil extraction kit following the manufacturer's protocol with minor modifications to increase the yield[53]. The V4–V5 region of the 16S rRNA gene was amplified (primers 515F/926R; Supplementary Table 1)[54,55] and prepared for sequencing following a standard Illumina Amplicon workflow[56]. Metagenomic shotgun libraries were prepared with the Illumina DNA Nextera Flex kit following the manufacturer's protocol. Marker gene sequences and shotgun metagenomics reads were sequenced on two Illumina MiSeq runs (reagent kit version 3, 600 cycles) at the Integrated Microbiome Resource facility at Dalhousie University.

**16S rRNA workflow.** Primer sequences were trimmed of forward and reverse primers using cutadapt[57]. Sequences were discarded when more than 12% of the bases in the primer sequence were incorrect. Remaining reads were further processed using DADA2[58] (v1.16.0) within the R environment[49]. Reads were dropped from the dataset if they had more than three expected errors (maxEE = 3), at least one base with very low quality (truncQ = 2), or at least one position with an unspecified nucleotide (maxN = 0). Based on visual inspection of quality plots, forward reads were trimmed to 260 bp and reverse reads 150 bp. Sequences were retained when both reads of a pair passed the filter. Remaining reads were dereplicated before inferring ASVs. Paired-end reads were merged and read pairs that did not match exactly across at least 12 bp were discarded. Merged reads were then screened for chimeras (method = consensus). Taxonomy was assigned to each ASV

using the naive Bayesian classifier[59] against the Silva_nr_v138_train_set reference database[60]. Prior to analysis, ASVs classified as Chloroplast, Mitochondria, or Eukaryota, or that remained unclassified (i.e., NA) at the kingdom level, were removed from the dataset using the package *phyloseq*[61] (v1.36.0) in the R environment[49].

Microbial ASVs that were enriched and more frequent in hypoxic and normoxic samples were identified using the Indicator Species Analysis (ISA)[62] computed with the R package *labdsv*[63] (v2.0-1). ASVs were considered an indicator of a group if they had a $p$ value less than or equal to 0.05. We used anvi'o[64] (v6.2-master) to visualize the distribution of all ASVs with more than 100 total reads (190 of 1272 total ASVs) and overlaid the results of the ISA. Any ASV that was found to be an indicator but had <100 reads was not included in the visualization. All 111 differentially expressed ASVs used in analyses are presented in Supplementary Data 1. Hierarchical clustering of both samples and ASVs were performed using Euclidean distance and ward linkage against the ASV/sample abundance matrix. Results from the ISA were included for each ASV in the visualization, as was Class level abundance for each sample.

**Metagenomic workflow.** We used Trimmomatic[65] (v0.39) for adapter clipping and initial quality trimming of raw metagenomic data ($N = 4$; one per site and time period). Within the anvi'o environment, we built a Snakemake[66] (v.5.10.0) workflow for co-assembly analysis. In the workflow, we used iu_filter_quality_minoche from the Illumina Utils package[67] (v2.11) for additional quality filtering and MEGAHIT[68] (v1.2.9) for co-assembly (–min-contig-len: 1000, –presets: meta-sensitive). Next, we used anvi-gen-contigs-database to generate a database of contigs and Prodigal[69] (v2.6.3) for gene calling. Within the workflow, KrakenUniq[70] (v0.5.8) was used for taxonomic classification of short reads against a user-constructed database of archaea, bacteria, viral, fungi, and protozoa reads from RefSeq and the NCBI nt database. Bowtie2 (v2.3.5) and SAMtools[71] (v1.10) were used to map short reads to the assembly and anvi-profile to create individual profile databases. We then used anvi-merge to generate a merged profile database of all four metagenome profiles. Taxonomic classification of contigs was performed using Kaiju[72] (v1.7.3, against the nr + euk database) and VirSorter[73] (v1.0.6, with the –virome flag and aligned using DIAMOND [74] v0.9.14).

For MAG reconstruction, we conducted an initial automatic binning step with CONCOCT[75] (v1.1.0), where the number of automatic bins was set to 5. We followed with a series of manual binning steps using anvi-refine to visually inspect and curate each bin by taking into account (1) GC content and taxonomy (based on Kaiju and VirSorter taxonomic profiles), (2) mean coverage and detection, (3) completion and redundancy assessment of bins based on the occurrence of single-copy genes (SCG), and (4) three independent scaffold clustering algorithms (differential coverage; tetra-nucleotide frequency; differential coverage and tetra-nucleotide frequency). We used Hidden Markov model (HMM) profiling of 76 SCG from Archaea[76], 71 SCG from Bacteria[77], 83 SCG for Protists (http://merenlab.org/Delmont-euk-scgs), and 12 ribosomal RNAs (https://github.com/tseemann/barrnap) to estimate the completion (C) and redundancy (R) of bins. Bins were considered a MAG if C was >70% and R <10%. Any bin where the majority of scaffolds were classified as viral was deemed a viral bin. The distribution of the bins across the four samples was visualized with the anvi'o interactive interface. Hierarchical clustering of bins was performed using Euclidean distance and Ward linkage against the relative abundance matrix. MAG taxonomy was assessed by running the command anvi-run-scg-taxonomy, which uses 22 ribosomal genes from the GTDB to estimate taxonomy[30].

Functional annotation of assembled contigs in the metagenome and individual MAGs were performed against KOfam (a customized HMM database of KEGG Orthologs[76]) using the command anvi-run-kegg-kofams within the anvi'o environment. We then used the command anvi-estimate-metabolism along with the KEGG annotations to reconstruct metabolic pathways (modules) and estimate pathway completeness[78]. Module completeness was defined as the proportion of steps that must be complete in order for the entire module to be marked complete. The module completeness threshold was set at 0.75.

**Phylogenomic analysis of MAG02 and MAG04.** A phylogenomic approach was used to further investigate the placement of MAG02 and MAG04 within the *Arcobacter* and *Alliiroseovarius* groups, respectively, because of their strict association with hypoxic conditions and their high frequency in the hypoxic 16S rRNA samples. All publicly available genomes classified in the *Arcobacter* group (NCBI TaxID 2321108) and *Aliiroseovarius* group (NCBI TaxID 1658781) were retrieved from the GenBank genome collections with the program ncbi-genome-download (v0.3.0, https://github.com/kblin/ncbi-genome-download). In total, 124 genomes classified as *Arcobacter* (including species formerly described as *Arcobacter*: *Aliiarcobacter, Haloarcobacter, Pseudoarcobacter, Poseidonibacter*) and 28 genomes classified in the *Aliiroseovarius* group (NCBI TaxID 1658781) were retrieved from the GenBank genome collections. We removed 22 *Arcobacter* genomes that had lower completion estimates than MAG02 (92.96% or <66 bacterial SCG) and/or no identifiable near full-length 16 s rRNA sequence. We also removed two *Aliiroseovarius* genomes that had lower completion estimates than MAG04 (83.1% or <59 bacterial SCG). We then used anvi-dereplicate-genomes and fastANI[79] to cluster genomes at 99% (including the MAGs). For *Arcobacter*, we selected a representative from each of the 73 clusters and added three *Sulfurospirillum*

(TaxID 5766) genomes as the outgroup for a final collection of 77 genomes. For *Aliiroseovarius*, we selected a representative from each of the 15 clusters and added four *Roseovarius* (TaxID 74030) genomes as the outgroup for a final collection of 19 genomes. We used the anvi'o pangenomic workflow to define gene clusters and the anvi'o interactive to screen SCG clusters that (1) were present in all 77 and 19 genomes, respectively, (2) occurred a maximum of one time in each genome, (3) had a maximum functional homogeneity index of 0.90 and 0.95, respectively, and (4) a minimum geometric homogeneity index of 0.99 and 1.00, respectively. This resulted in a collection of 17 gene clusters (1309 genes) for the *Arcobacter* (plus *Sulfurospirillum* outgroup) analysis, and 28 gene clusters (532 genes) for the *Aliiroseovarius* (plus *Roseovarius* outgroup) analysis. The command anvi-get-sequences-for-gene-clusters was used to retrieve, align (performed using MUSCLE[80] v3.8.1551), and concatenate target gene sequences for each genome. IQ-TREE 2[81] (v2.0.3) and ModelFinder Plus[82] identified LG+F+R4 and LG+F+I +G4 as the most suitable substitution models (out of 48 protein models tested) based on Bayesian information criterion (BIC) for *Arcobacter* (plus *Sulfurospirillum* outgroup), and *Aliiroseovarius* (plus *Roseovarius* outgroup), respectively. We subsequently ran these models with 5000 ultrafast bootstrap approximations with UFBoot2[83] on the curated collection of genomes and SCG.

Habitat fidelity of relatives of MAG02 and MAG04 was evaluated based on metadata from each genome in NCBI's BioSample database. Each genome was assigned to a broad habitat category (vertebrate host, invertebrate host, marine surface, marine subsurface, fuel cell, and sewage) based on the host and isolation source (one genome had no information on isolation source or host). Vertebrate hosts almost entirely encompassed terrestrial animals such as cow, duck, human, and pig, and isolations sources like aborted fetus (not human), blood, eye, skin, and slaughterhouse. Invertebrate hosts were exclusively marine and included mussel, oyster, abalone, scallop, and clam. Marine surface sources included seawater and surface seawater, while marine subsurface sources included sediment and marine subsurface aquifer. Fuel cells were isolates obtained from fuel cell enrichment. The sewage category sources were of terrestrial origin and also included wastewater and reclaimed water. See Supplementary Data 3 and Supplementary Data 4 for metadata records of all genomes used in the *Arcobacter* and *Aliiroseovarius* phylogenomic analyses, respectively.

**Reporting summary**. Further information on research design is available in the Nature Research Reporting Summary linked to this article.

## Data availability

Trimmed 16S rRNA (primers removed) sequence data generated in this study are deposited in the European Nucleotide Archive (ENA) under Project Accession number PRJEB36632 (ERP119845), sample accession numbers ERS4291994-ERS4292031. Raw 16S rRNA fastq files can be accessed through the Smithsonian figshare, https://doi.org/ 10.25573/data.11819745. The metagenomic sequence data generated in this study are deposited in the ENA under Project Accession number PRJEB36632 (ERP119845), sample accession numbers ERS4578390-ERS4578393. Related data and data products for individual analysis workflows are available through the Smithsonian figshare under the collection https://doi.org/10.25573/data.c.5025362 [https://doi.org/10.25573/data. c.5025362.v1]. Source data are provided with this paper.

## Code availability

All code, reproducible workflows, and further information on data availability can be found on the project website at https://hypocolypse.github.io/. The code embedded in the website is available on GitHub [https://github.com/hypocolypse/web] in R Markdown format. The version of code used in this study is archived under Hypocolypse Workflows v1.0[84] [https://github.com/hypocolypse/web/releases/tag/v1.0], DOI identifier, https:// doi.org/10.5281/zenodo.4940132.

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

## Acknowledgements

M.D.J. was funded by postdoctoral fellow awards from the Smithsonian Institution's Marine Global Earth Observatory (MarineGEO) and the Smithsonian Tropical Research Institute (STRI); M.L. and N.L. were funded by postdoctoral support from the STRI Office of Fellowships. J.J.S. was funded by a grant from the Gordon and Betty Moore Foundation awarded to STRI and UC Davis (doi:10.37807/GBMF5603). L.M.R.B., W.L.W., and A.H.A. were supported by MarineGEO, a private funder, and STRI funds to A.H.A. Many of the computations were conducted on the Smithsonian High-Performance Cluster (SI/HPC), Smithsonian Institution (doi:10.25572/SIHPC). We thank Rachel Collin for facilities support at the Bocas del Toro Research Station, Plinio Gondola and the research station staff for logistical support, Roman Barco for insight into the functional analyses, Sherly Castro for informative feedback, and Mike Fox for assistance with community analyses. Research permits were provided by the Autoridad Nacional del Ambiente de Panamá. This paper is the result of research funded by the National Oceanic and Atmospheric Administration's National Centers for Coastal Ocean Science Competitive Research Program under award NA18NOS4780170 to A.H.A. and M.D.J. through the University of Florida. This is contribution 257 from the Coastal Hypoxia Research Program and 86 from the Smithsonian's MarineGEO and Tennenbaum Marine Observatories Network.

## Author contributions

M.D.J., J.J.S., M.L., N.L., and A.H.A. conceived and designed the study. M.D.J., N.L., L.M.R.B., and W.L.W. conducted surveys and collected field samples. N.L., L.M.R.B., and W.L.W. conducted laboratory analyses. J.J.S. and M.L. performed extractions and sample processing for sequencing. M.D.J., J.J.S., M.L., and N.L. conducted statistical analyses and created figures. M.D.J. and J.J.S. drafted the manuscript and supporting information, and managed revisions with input from all co-authors.

## Competing interests

The authors declare no competing interests.
