## [Peer Review File · Nature Communications]

Reviewers' Comments:

Reviewer #1:

Remarks to the Author:

General comments about the manuscript:

The consequences of human activities on coastal ecosystems are extensive across all marine habitats, though coral reefs exhibit degradation at the global scale because of the numerous negatively impact their survival. Beyond heat stress, nutrient enrichment and the reduction of algal grazers through fishing activities, acute deoxygenation is becoming established as an emerging stressor that diminishes reef water quality and causes mass mortality on coral reefs. The study described in this manuscript characterizes the underappreciated environmental impact of acute deoxygenation occurring at a Caribbean reef system. In the article, descriptions of reef-scale shifts in benthic community composition were coupled with an analysis of reef microbes during and after the hypoxic event at impacted and control sites. I previously reviewed an earlier version of this manuscript and I commend the authors for their revisions and particularly appreciated the inclusion of more details on the microbial analyses. The field needs more studies that incorporate microbial dynamics into investigations of climatic events which impact the health of complex habitats. The study also highlights the need to further investigate the occurrence of deoxygenation during warming events as this could be a contributing factor in coral bleaching.

The text is well written, the figures are sophisticated, informative, and clear. For example, the introduction conveys important details for a broad audience about deoxygenation events occurring at a global scale and effectively summarizes the state of coral reefs. The statistical analyses are strong, and the conclusions are well justified by the data. The article is extremely well written, using active language to develop a cohesive argument. The scientific findings will be interesting to a broad audience, supporting publication in Nature Communications. I also support the journal choice for accessibility to researchers and managers across the world.

Below are a couple specific recommendations for minor revisions:

Line 244, differ  different

The Poseidonibacter versus Arcobacter is still a bit unclear. Figure 3 reports MAG02 as Poseidonibacter, but discuss text 268 – 281 mainly refers to MAG02 as Arcobacter. It was difficult for me to decipher why that was.

Figure 3 legend. Line 305 ward  Ward.

The legend mentions 5 MAGs but there appear to be 6 with the inclusion of Nitrospiraceae.

Reviewer #2:

Remarks to the Author:

Overall

This study describes coral reef community responses to a hypoxic event, including the benthic macro-organisms to the microbial communities in the overlying seawater. This is an important and novel study that highlights the drastic ecological consequences of brief hypoxic events on benthic coral reef communities, and the comprehensive analyses of both the benthic community composition with the microbial communities is informative as it highlights the decoupling of the recovery trajectories of these two different communities. The discussion would be improved by better integrating the interpretation of the benthic vs. microbial community results. In addition, some of the discussion needs more detail and evidence to support the author's arguments. Specifically, this includes the comparison of the relative threats of hypoxic vs. thermal coral bleaching events. Thermal bleaching is expected to become annual in the next few decades, and can also lead to near complete mortality of some coral communities (see GBR 2015-16), but the authors do not discuss how often or widespread hypoxic events are despite claiming they may pose a more severe local threat than warming. In addition, the local management implications or imperatives are brought up but no details on the role of anthropogenic factors driving hypoxia or

its mitigation are mentioned.

Specific comments

“control” site should be changed to “reference” site throughout.

L83: state how many days, or if not known the minimum number of days it was sustained

L84-85: Better to introduce previous hypoxic event (drivers and consequences) in the introduction. Also please state the depth range for mass mortality in that event, not just the upper reach.

L97-98: this sentence is not relevant here, suggest moving to discussion

L93-102: the coral descriptions and hypoxia responses should go in the coral response section

L140-1: be transparent and define substantial with % or fold difference or actual values. Also, it is not surprising that any of the other variables do not differ as much as DO, since this is a hypoxia event, but that does not mean the other smaller differences are not significant or biologically meaningful.

L230-31: What evidence supports this claim? What were the ‘marked differences’? The comparison between the two events is confusing and not well described. If all the corals died in the earlier event, of course recovery of coral would be slower. Also, did that happen because the event was more severe that year? Or was it due to species specific differences in hypoxia tolerance as you are implying? There needs to be evidence to support these claims, right not this discussion is very speculative and vague.

Figures: these are all beautifully done

Fig1c: I can’t see the arrow very well or the microbial mats it is pointing to

Discussion

Need to integrate microbial responses into the first paragraph, and connect benthic communities to microbial communities throughout this section. For example, what implications do the hypoxia-associated microbes have for benthic organisms (corals, other inverts)?

L419: how often do these events occur, in this location and reefs around the world?

424: How would local policies influence the occurrence of hypoxic events? What human factors are driving these events, and what types of policies would be helpful?

Reviewer #3:

Remarks to the Author:

The manuscript entitled “Rapid ecosystem-scale consequences of acute deoxygenation on a Caribbean coral reef” describes abiotic conditions and biotic community responses during a hypoxic event and after a year-long recovery period. This work is critically important to understanding ongoing hypoxic stress to reef ecosystems. The experimental design is sound and captures important timepoints before and during the hypoxic event. The authors find that an acute hypoxic event caused a dramatic shift in benthic community structure up to one year later, including a 50% decrease in live coral cover. Microbial communities were similarly impacted but returned to pre-hypoxic states within a month. I defer to a microbiologist’s expertise in reviewing the authors’ conclusions with respect to exact microbial taxa and molecular functions associated with the hypoxic event. I find their genetic analyses convincing. I am excited to see this work published. Minor comments below.

Line 25: Please clarify the time scale referenced when describing the “unprecedented rate” of

declines in dissolved oxygen. E.g., is the current rate unprecedented in the 500 million years since coral species have existed or some more modern era?

Line 50: remove extra "has"

Line 83: maybe a missing number before the word "days" (e.g., "three days long")? Otherwise, should the sentence read "A day long...?"

Figure 1: I found it difficult to follow the figure as the subpanel and legend sequence does not proceed alphabetically. I suggest rearranging the figure and legend to follow the logical sequence.

Line 133 in Figure A legend: I do not see white arrows in subpanel (a) that is being described by this sentence.

Line 137: I recommend describing the brittle stars' behaviors more generally, without ascribing intention to their actions. For example, "brittle stars congregating on the tops of mounding corals."

Figure 3: Wow! Panel A has so much data displayed so beautifully. The legend appears to be missing instructions for how to interpret the grey bars between the heatmap and taxonomic profiles (i.e., no. reads, no. asvs, Shannon, and Inverse Simpson). That information is the hardest to read too. The authors could consider presenting this information as a new figure/panel/supplement but I could go either way if it's described in the legend. WOW again for Panel B. The symbols/text for each layer of the individual MAGs is too small to read. Again- perhaps this could be another figure/supplement.

Figure 4: Another beautiful figure. Here the proportion of coral habitat bar graphs don't provide tremendous support for the statement in the legend that "the deoxygenation event precipitated ... a change in the benthic assemblage that persisted" because we don't see the "before" estimates of coral cover like was shown in Figure 2b. I'm assume those data are not shown here because the authors do not have microbial data to show from before. Perhaps the authors could present the proportion of coral habitat as a change from the "before" observation for both sites to better demonstrate these differences.

Reviewer #1

General Comments:

The consequences of human activities on coastal ecosystems are extensive across all marine habitats, though coral reefs exhibit degradation at the global scale because of the numerous negatively impact their survival. Beyond heat stress, nutrient enrichment and the reduction of algal grazers through fishing activities, acute deoxygenation is becoming established as an emerging stressor that diminishes reef water quality and causes mass mortality on coral reefs. The study described in this manuscript characterizes the underappreciated environmental impact of acute deoxygenation occurring at a Caribbean reef system. In the article, descriptions of reef-scale shifts in benthic community composition were coupled with an analysis of reef microbes during and after the hypoxic event at impacted and control sites. I previously reviewed an earlier version of this manuscript and I commend the authors for their revisions and particularly appreciated the inclusion of more details on the microbial analyses. The field needs more studies that incorporate microbial dynamics into investigations of climatic events which impact the health of complex habitats. The study also highlights the need to further investigate the occurrence of deoxygenation during warming events as this could be a contributing factor in coral bleaching.

The text is well written, the figures are sophisticated, informative, and clear. For example, the introduction conveys important details for a broad audience about deoxygenation events occurring at a global scale and effectively summarizes the state of coral reefs. The statistical analyses are strong, and the conclusions are well justified by the data. The article is extremely well written, using active language to develop a cohesive argument. The scientific findings will be interesting to a broad audience, supporting publication in Nature Communications. I also support the journal choice for accessibility to researchers and managers across the world.

Response

We thank Reviewer 1 for their encouraging words, constructive feedback, and continued support for this manuscript.

Specific Comments:

Comment 1-1

Below are a couple specific recommendations for minor revisions:

Line 244, differ  different

Response 1-1

We have edited the text accordingly (Line 308).

Comment 1-2

The Poseidonibacter versus Arcobacter is still a bit unclear. Figure 3 reports MAG02 as Poseidonibacter, but discuss text 268 – 281 mainly refers to MAG02 as Arcobacter. It was difficult for me to decipher why that was.

Response 1-2

The taxonomy for *Arcobacter* is contentious and in flux. We have added text to acknowledge the issues with this taxonomy, and refer to this taxon as *Arcobacter* throughout the manuscript. For clarity, we have updated the figure text for MAG02 to *Arcobacter* (rather than *Poseidonibacter*).

Line 233-235: *MAG02, an Arcobacter (Arcobacteraceae, Campylobacterota), and MAG04, an Alliroseovarius (Rhodobacteraceae, Alphaproteobacteria), were found exclusively in the hypoxic sample (Fig. 3b).*

Line 237-241: *It is worth noting that the taxonomy of Arcobacter is currently in flux²⁸ and Pérez-Cataluña et al. (2018) recently proposed splitting Arcobacter into six genera²⁹. Though taxonomic analysis against the Genome Taxonomy Database (GTDB)³⁰ (see Methods) classified MAG02 as a Poseidonibacter, for clarity, we refer to MAG02 as Arcobacter throughout.*

Line 391-394: *Notably, we identified an Arcobacter (Arcobacteraceae) (MAG02) and an Alliroseovarius (Rhodobacteraceae) (MAG04) with marine origins that were both dominant and unique to hypoxic conditions, but were absent from an analogous assemblage collected at the same site a month later, following the return of normoxic conditions.*

Comment 1-3

Figure 3 legend. Line 305 ward  Ward.

Response 1-3

We have edited the text accordingly (Line 280).

Comment 1-4

The legend mentions 5 MAGs but there appear to be 6 with the inclusion of Nitrospiraceae.

Response 1-4

We have clarified the treatment of Nitrospiraceae in the text, and figure legend accordingly, to indicate that it was not fully resolved as a MAG and is referred to as Bin 13 instead.

Line 235-237: *A sixth abundant microbial bin (Bin 13, unclassified Nitrospiraceae) was found exclusively in the hypoxic sample. Despite manual bin curation, Bin 13 maintained high redundancy (68%) and could not be confidently resolved to a high-quality MAG (Fig. 3b).*

Line 291-293: (Figure Legend 3b) *The five MAGs recovered from the assembly are overlaid, including genus-level taxonomic assignments. Bin 13 (Nitrospiraceae), an unresolved bin, is also included because it was abundant and only detected in the hypoxic samples.*

Reviewer #2

General Comments:

This study describes coral reef community responses to a hypoxic event, including the benthic macro-organisms to the microbial communities in the overlying seawater. This is an important and novel study that highlights the drastic ecological consequences of brief hypoxic events on benthic coral reef communities, and the comprehensive analyses of both the benthic community composition with the microbial communities is informative as it highlights the decoupling of the recovery trajectories of these two different communities. The discussion would be improved by better integrating the interpretation of the benthic vs. microbial community results. In addition, some of the discussion needs more detail and evidence to support the author's arguments. Specifically, this includes the comparison of the relative threats of hypoxic vs. thermal coral bleaching events. Thermal bleaching is expected to become annual in the next few decades, and can also lead to near complete mortality of some coral communities (see GBR 2015-16), but the authors do not discuss how often or widespread hypoxic events are despite claiming they may pose a more severe local threat than warming. In addition, the local management implications or imperatives are brought up but no details on the role of anthropogenic factors driving hypoxia or its mitigation are mentioned.

Response

We thank Reviewer 2 for their careful and constructive review of our manuscript, and for the insightful comments on how to improve our interpretations and discussion of implications. We have integrated our benthic and microbial results in the first paragraph of the discussion and throughout as suggested (Lines 323-340 & 387-391), and expanded (and toned down) our discussion of hypoxia versus thermal bleaching events (Line 421-425). We also include more discussion about specific management strategies (Line 425-433).

Specific Comments:

Comment 2-1

“control” site should be changed to “reference” site throughout.

Response 2-1

Thank you for this suggestion. We have updated the text and swapped “reference site” in place of “control site” throughout.

Comment 2-2

L83: state how many days, or if not known the minimum number of days it was sustained

Response 2-2

We have edited the text accordingly.

Line 97-98: *Although the exact duration of the event is unknown, hypoxic conditions were detected on the shallow reef over a period spanning at least six days (Sept 20-26, 2017).*

Comment 2-3

L84-85: Better to introduce previous hypoxic event (drivers and consequences) in the introduction. Also please state the depth range for mass mortality in that event, not just the upper reach.

Response 2-3

We have moved discussion of the drivers of hypoxic events to the introduction, as well as discussion of the 2010 event. We include the depth range for the mass mortality event from 2010, from 10-12 m depth to the bottom of the reef (~15 m), rather than >10-12 m.

Line 72-77: *The first well-documented assessment of hypoxia on a coral reef followed an event in late September, 2010 in Bocas del Toro on the Caribbean coast of Panama⁴. Formation of hypoxic conditions was likely related to a period of low wind activity, warmer water temperatures corresponding to the end of the boreal summer, and high levels of eutrophication and organic pollution in Bahía Almirante⁴. The 2010 event resulted in mass mortality of virtually all corals at a depth from 10-12 m down to the bottom of reefs at ~15 m at affected sites⁴.*

Comment 2-4

L97-98: this sentence is not relevant here, suggest moving to discussion

Response 2-4

We agree that this sentence is out of context, and have instead streamlined all discussion of composition of the shallow versus deeper reef communities (and their response to a deoxygenation event) to the discussion. See **Response 2-7** for more details.

Comment 2-5

L93-102: the coral descriptions and hypoxia responses should go in the coral response section

Response 2-5

We agree with this suggestion, and have moved the specific descriptions of the coral community to the coral response section. We believe including our qualitative observations during the event is important, which includes our general description of coral bleaching during the event and impacts on other benthic taxa (as seen in Figure 1). We have changed the section subheading to “Characteristics of a deoxygenation event” to encompass both our *in situ* observations during the event and our measurements of environmental conditions.

Line 87-98: ***Characteristics of an acute deoxygenation event***

An acute deoxygenation event occurred again in Bahía Almirante on the Caribbean coast of Panama (Fig. 1) in 2017, seven years after the first reported hypoxic event in the area⁴. A distinct sharp gradient of DO, or oxycline, was first detected in late September on a representative inner-bay coral reef (Cayo Roldan, hereafter impacted site) at 3-4 m depth (Fig. 1d), which likely resulted from the shoaling of hypoxic water that can persist in the water column adjacent to reef habitats at depths below 20 m in Bahía Almirante¹⁵. Below this shallow oxycline, the benthos was littered with dead, but intact sponges and exposed, moribund, and dying mobile

invertebrates (Fig. 1e), as well as hypoxia-associated microbial mats (Fig. 1b). Coral bleaching, tissue loss, and mortality were apparent and widespread within the hypoxic water (Fig. 1a, 1c). Although the exact duration of the event is unknown, hypoxic conditions were detected on the shallow reef over a period spanning at least six days (Sept 20-26, 2017).

Comment 2-6

L140-1: be transparent and define substantial with % or fold difference or actual values. Also, it is not surprising that any of the other variables do not differ as much as DO, since this is a hypoxia event, but that does not mean the other smaller differences are not significant or biologically meaningful.

Response 2-6

We agree with this suggestion, and to add transparency we have updated the text to include the exact range in pH, temperature, salinity, and chlorophyll between the two sites during the event, in addition to the range in DO.

Line 109-114. *The difference in environmental conditions between sites was greatest for DO, with a maximum range in concentrations of 6.57 mg L⁻¹. Other parameters also varied between the focal sites, but the maximum range between sites was minimal for pH (0.72 units), temperature (1.7 °C), salinity (2.27 practical salinity units, PSU), and chlorophyll (2.07 relative fluorescent units, RFU) relative to the changes in DO (Supplemental Fig. 1).*

Comment 2-7

L230-31: What evidence supports this claim? What were the ‘marked differences’? The comparison between the two events is confusing and not well described. If all the corals died in the earlier event, of course recovery of coral would be slower. Also, did that happen because the event was more severe that year? Or was it due to species specific differences in hypoxia tolerance as you are implying? There needs to be evidence to support these claims, right not this discussion is very speculative and vague.

Response 2-7

Thank you for this suggestion. We agree and have streamlined our discussion of the shallow versus deeper reef communities into one paragraph in the discussion (also to address **Comment 2-4**). To support the statement for higher resilience and recovery potential, we include specific examples from the present study as well as examples from the 2010 observations on nearby reefs.

Line 341-344 & 358-377. *The shallow reef benthic community appeared to be more resilient to an acute deoxygenation event than the adjacent deeper reef coral community, and may show a greater potential for recovery. We observed that virtually all corals in the deeper reef community (at depths ranging from 10–20 m) suffered rapid mortality in 2017 (M. Johnson, personal observation). Yet, we found that nearly half of living corals in the shallow reef community survived. During the 2010 event, nearly all corals at and below 10-12 m died on nearby reefs, but the shallow coral reef adjacent to those communities was not affected. While the deeper reef*

community failed to recover from the 2010 event, the shallow reef community affected by the 2017 event showed signs of some recovery one year later. Variation in the response of different reef assemblages to an acute deoxygenation event could be related to stress-tolerances of the dominant species in each assemblage. In Bahía Almirante, the reefs that persist along the 3-5 m versus 10-12 m depth profiles vary fundamentally in species composition and abundance. The less diverse shallow assemblage is dominated by more stress-tolerant corals, including Porites and Millepora, while the deeper assemblage is dominated by boulder corals (e.g., Orbicella, Siderastrea, and Montastrea) and plating corals (Agaricia lamarcki)^{23,37}. Within the shallow community studied here, Agaricia appeared to be more negatively affected than the other abundant corals. Greater hypoxia-sensitivity has been documented for another species of Agaricia (A. lamarcki) relative to other corals found on deeper reefs. Such differential tolerances can result in a shift towards assemblages composed of more tolerant species rather than outright loss of the coral community following a hypoxic event⁴, which may have enabled higher survival in the shallow reef community during the deoxygenation event and facilitated gradual recovery a year later. Higher survival of coral taxa in shallower environments following these events is critical for reef recovery dynamics, because surviving corals may provide a source of larvae to facilitate recolonization and recovery of the deeper reef community³⁸.

Comment 2-8

Figures: these are all beautifully done

Response 2-8

Thank you!

Comment 2-9

Fig1c: I can't see the arrow very well or the microbial mats it is pointing to

Response 2-9

We have moved the arrow so it is more visible, and included a physical description of the mats in the figure legend to help with visualization.

Line 119-122. *Below the oxycline in hypoxic waters, (b) microbial mats diagnostic of anoxic conditions, comprised of thin, white filaments (red arrow), were abundant, and (c) coral bleaching typically progressed from the bottom to the top of boulder corals (white arrow).*

Comment 2-10

(Discussion) Need to integrate microbial responses into the first paragraph, and connect benthic communities to microbial communities throughout this section. For example, what implications do the hypoxia-associated microbes have for benthic organisms (corals, other inverts)?

Response 2-10

Thank you for this suggestion, we have integrated the microbial responses into the discussion, particularly in the first paragraph, and further connected the benthic and microbial communities.

Little is known about the microbial taxa that were found exclusively in the hypoxic waters, particularly their effects on other marine taxa. Because of the lack of information, we don't know what the implications of those microbes are for the benthic organisms, other than providing further support for the presence of hypoxic-anoxic waters. We have focused on what is known about potential impacts/roles of microbes on benthic processes. To link the microbial community assemblages to the benthic assemblages, we discuss how the rapid return to normoxic microbial assemblages may allow for recovery of the benthic reef assemblage by facilitating resumption of key microbial processes.

Line 323-340: *Here we captured the first micro-to-macroscopic responses of a shallow community as an acute deoxygenation event unfolded on a Caribbean coral reef (Fig. 4). Conditions associated with the hypoxic event coincided with bleaching, tissue loss, and whole colony mortality in corals at the impacted site. These findings were evident in both physiological measurements collected during the event and visual surveys of coral bleaching and mortality before, during, and after the event. When bleaching is detected on a coral reef, thermal stress is the commonly assumed culprit, and much of our understanding of bleaching in corals has focused on temperature. Our study suggests that the role of oxygen-stress in the field also should be considered in future studies, and supports emerging lab-based evidence that hypoxia can initiate a coral bleaching response²⁰. An increase in dead and bleached corals during the event corresponded with a shift in the microbial assemblage to distinct taxa that thrived in hypoxic conditions (e.g., Arcobacter). But unlike the macrobenthos, the microbial assemblage rapidly reverted to a normoxic assemblage within a month of the event. Though the number of studies to implicate hypoxia in mass mortalities on coral reefs is increasing^{16,36}, ours is the first to characterize a coral bleaching response to in situ conditions associated with an active deoxygenation event and the concurrent responses of the benthic and microbial assemblages. Furthermore, no studies have previously described the consequences of hypoxic waters shoaling to such shallow depths on a coral reef.*

Line 387-391: *Further, the shift we detected from the hypoxic microbial community to a normoxic assemblage after the event subsided suggests that the recovery trajectory of reef-associated water-borne microbes is independent and decoupled from the benthic macrofauna. This may facilitate the resumption of key microbial processes (e.g., biogeochemical cycling) that influence the recovery of other aspects of the reef community.*

Comment 2-11

L419: how often do these events occur, in this location and reefs around the world?

Response 2-11

We have revised the text with more information about the occurrence of hypoxia events at this site, and what limited information exists for other tropical locations.

Line 421-425: *While the frequency of bleaching events is well-understood, less is known about the occurrence of hypoxic events in the tropics. The event we document here is the second major event in Bocas del Toro⁴, and adds to ~50 known mass mortality events associated with acute*

hypoxia in the tropics globally, though that number is likely underestimated by an order of magnitude⁴.

Comment 2-12

424: How would local policies influence the occurrence of hypoxic events? What human factors are driving these events, and what types of policies would be helpful?

Response 2-12

We have included more in the discussion about local policies and the occurrence of hypoxia events, and present drivers of the hypoxia events for the region in the introduction.

Line 425-433: *Key steps towards improving our understanding of the environmental characteristics and ecosystem implications of hypoxic events are to identify habitats that are at high risk⁴ and to incorporate regular and continuous monitoring of DO concentrations^{14,36}. Their localized nature suggests that policy and management practices, including managing watershed inputs like eutrophication and raw waste effluent, may be effective in reducing the likelihood that a hypoxia event will occur³. Coral reefs are already suffering severe losses in response to a range of environmental stressors, and our study illustrates the importance of incorporating deoxygenation into experimental frameworks and management plans.*

Reviewer #3

General Comments:

The manuscript entitled “Rapid ecosystem-scale consequences of acute deoxygenation on a Caribbean coral reef” describes abiotic conditions and biotic community responses during a hypoxic event and after a year-long recovery period. This work is critically important to understanding ongoing hypoxic stress to reef ecosystems. The experimental design is sound and captures important timepoints before and during the hypoxic event. The authors find that an acute hypoxic event caused a dramatic shift in benthic community structure up to one year later, including a 50% decrease in live coral cover. Microbial communities were similarly impacted but returned to pre-hypoxic states within a month. I defer to a microbiologist’s expertise in reviewing the authors’ conclusions with respect to exact microbial taxa and molecular functions associated with the hypoxic event. I find their genetic analyses convincing. I am excited to see this work published. Minor comments below.

Response

Thank you for the insight and feedback. We appreciate the positive assessment of our work.

Specific Comments:

Comment 3-1

Line 25: Please clarify the time scale referenced when describing the “unprecedented rate” of declines in dissolved oxygen. E.g., is the current rate unprecedented in the 500 million years since coral species have existed or some more modern era?

Response 3-1

We have edited the text to remove the reference to “unprecedented rate”, and instead refer to acceleration oxygen loss in the global ocean.

Line 25-26: *Loss of oxygen in the global ocean is accelerating due to climate change and eutrophication, but how acute deoxygenation events affect tropical marine ecosystems remains poorly understood.*

Comment 3-2

Line 50: remove extra “has”

Response 3-2

We have edited the text accordingly (Line 50).

Comment 3-3

Line 83: maybe a missing number before the word “days” (e.g., “three days long”)? Otherwise, should the sentence read “A day long...”?

Response 3-3

We have edited the text accordingly.

Line 88-89: *An acute deoxygenation event occurred again in Bahía Almirante on the Caribbean coast of Panama (Fig. 1) in 2017, seven years after the first reported hypoxic event in the area⁴.*

Line 97-98: *Although the exact duration of the event is unknown, hypoxic conditions were detected on the shallow reef over a period spanning at least six days (Sept 20-26, 2017).*

Comment 3-4

Figure 1: I found it difficult to follow the figure as the subpanel and legend sequence does not proceed alphabetically. I suggest rearranging the figure and legend to follow the logical sequence.

Response 3-4

We have relabeled the panel figures in alphabetical order and made the corresponding changes to the figure legend.

Line 116-126: ***Fig. 1 / Dissolved oxygen concentrations and responses of coral reef taxa during the 2017 deoxygenation event in Bocas del Toro, Panama.*** (a) Coral bleaching (white arrow) was evident in corals at the impacted site (Cayo Roldan, white triangle) during the 2017 event, including the lettuce coral *Agaricia tenuifolia*. Below the oxycline in hypoxic waters, (b) microbial mats diagnostic of anoxic conditions, comprised of thin, white filaments (red arrow), were abundant, and (c) coral bleaching typically progressed from the bottom to the top of boulder corals (white arrow). (d) Map of Bahía Almirante illustrates DO concentrations at the

benthos six days after initial detection of the event (depths ~3-20 m), with sampling stations noted as black dots. An unaffected coral reef (Cayo Coral, black triangle) located outside Bahía Almirante was used as a reference site. (e) Below the oxycline at the impacted site, invertebrates, including cryptic brittle stars, congregated on the tops of boulder corals.

Comment 3-5

Line 133 in Figure A legend: I do not see white arrows in subpanel (a) that is being described by this sentence.

Response 3-5

We have moved the arrow in the panel to improve visibility.

Comment 3-6

Line 137: I recommend describing the brittle stars' behaviors more generally, without ascribing intention to their actions. For example, "brittle stars congregating on the tops of mounding corals."

Response 3-6

We have incorporated this suggestion into the revised Figure 1 legend.

Line 125-126: *(e) Below the oxycline at the impacted site, invertebrates, including cryptic brittle stars, congregated on the tops of boulder corals.*

Comment 3-7

Figure 3: Wow! Panel A has so much data displayed so beautifully. The legend appears to be missing instructions for how to interpret the grey bars between the heatmap and taxonomic profiles (i.e., no. reads, no. asvs, Shannon, and Inverse Simpson). That information is the hardest to read too. The authors could consider presenting this information as a new figure/panel/supplement but I could go either way if it's described in the legend. WOW again for Panel B. The symbols/text for each layer of the individual MAGs is too small to read. Again- perhaps this could be another figure/supplement.

Response 3-7

Thank you for this suggestion, and for the positive feedback. We have opted to keep the figure as panels a and b, and incorporated the suggestions for improving visibility of different components. Maintaining the panel format, versus separate figures, is key because we can make the direct link between the MAGs and the corresponding ASVs using arrows (thus connecting the data sets). We have made the distinction between panel a and b clearer, and have adjusted the diversity metrics in panel a so the text is bigger and the specific diversity metrics are clear. These have also been colored to match the sample color coding (blue for normoxic, vermilion for hypoxic). For panel b, we have removed the text that is ancillary, which was previously difficult to see. (See revised Figure 3 below)

Comment 3-8

Figure 4: Another beautiful figure. Here the proportion of coral habitat bar graphs don't provide tremendous support for the statement in the legend that "the deoxygenation event precipitated ... a change in the benthic assemblage that persisted" because we don't see the "before" estimates of coral cover like was shown in Figure 2b. I'm assume those data are not shown here because the authors do not have microbial data to show from before. Perhaps the authors could present the proportion of coral habitat as a change from the "before" observation for both sites to better demonstrate these differences.

Response 3-8

Thank you for this suggestion regarding clarification of the figure. We have edited the text to better reflect and explain the patterns shown, and have clarified that the coral (dead, live, bleached) proportion bars are the proportion of the coral habitat at each given time point that were either dead, live, or bleached. For this comparison, we focused on the coral and microbial community "during" and "after" the event for the periods where we had both coral and microbial data. We agree that understanding the changes in the coral community "during" and "after" the event relative to before is highly informative, and refer to Figure 2b-e where the magnitude of those differences through time within the coral community are presented.

Line 346-356: ***Fig. 4 | Impacts of an acute deoxygenation event on coral reef benthic and microbial community structure.*** Conceptual figure depicting how the deoxygenation event shifted microbial and benthic community composition, and how the shift in the benthic assemblage persisted one year after the event. In contrast, no such changes were observed at the reference site. Abundance of microbial taxa depicted in the water is proportional to the profiles determined through shotgun metagenome sequencing (top, horizontal stacked bars). The amount of coral habitat is presented as the proportion of coral habitat that was either live (brown), bleached (gray), or dead (dark gray) at each time point based on field surveys (bottom, horizontal stacked bars).

Reviewers' Comments:

Reviewer #2:

Remarks to the Author:

Great, work. The reviewers' comments have been sufficiently addressed and I look forward to seeing this study published.

Reviewer #3:

Remarks to the Author:

The reviewers addressed all of my comments. Great work.